# *Pythium banihashemianum sp. nov.* and *Globisporangium izadpanahii sp. nov.*: Two New Oomycete Species from Rice Paddies in Iran

**DOI:** 10.3390/jof10060405

**Published:** 2024-06-05

**Authors:** Fatemeh Salmaninezhad, Reza Mostowfizadeh-Ghalamfarsa, Santa Olga Cacciola

**Affiliations:** 1Department of Plant Protection, School of Agriculture, Shiraz University, Shiraz 7144167186, Iran; f.salmaninezhad@shirazu.ac.ir; 2Department of Agriculture, Food and Environment (Di3A), University of Catania, 95123 Catania, Italy

**Keywords:** multigene phylogenetic analysis, *Pythium*, *Oryza sativa*, pathogenicity, root and crown rot, taxonomy

## Abstract

An investigation into oomycete diversity in rice paddies of Fars Province in Iran led to the identification of two new *Pythium sensu lato* (*s.l.*) species as *Globisporangium izadpanahii sp. nov.* and *Pythium banihashemianum sp. nov.* The identification was based on morphological and physiological features as well as on the phylogenetic analysis of nuclear (ITS and *βtub*) and mitochondrial (*cox*1 and *cox*2) loci using Bayesian inference and Maximum Likelihood. The present paper formally describes these two new species and defines their phylogenetic relationships with other congeneric species. According to multiple gene genealogy analysis, *G. izadpanahii sp. nov.* was grouped with other species of *Globisporangium* (formerly, clade G of *Pythium s.l.*) and was closely related to both *G. nagaii* and the recently described *G. coniferarum.* The second species, designated *P. banihashemianum sp. nov.*, was grouped with other species of *Pythium sensu stricto* (formerly, clade B of *Pythium s.l.*) and, according to the phylogenetic analysis, shared an ancestor with *P. plurisporium.* The production of globose hyphal swellings was a major characteristic of *G. izadpanahii sp. nov.*, which did not produce vesicles and zoospores. In pathogenicity tests on rice seedlings, *P. banihashemianum sp. nov.* isolates were highly pathogenic and caused severe root and crown rot, while *G. izadpanahii sp. nov.* isolates were not pathogenic.

## 1. Introduction

*Pythium sensu lato* (*s.l.*) Pringsh. is a cosmopolitan, morphologically and genetically heterogeneous oomycete genus comprising more than 230 described species [1]. Several species of this genus have been reported as both facultative saprobes and plant, animal, and human pathogens [2,3,4,5,6,7,8]. Many other species have been reported as exclusively saprobes or even beneficial antagonists of plant pathogens [9,10,11,12,13,14,15]. Plant pathogenic species cause pre- and post-emergence damping off of seedlings, resulting in seed rot or the death of seedlings prior to emergence or the death of seedlings following emergence, respectively, as well as crown and root rot and may have a highly destructive impact on crops.

In the light of advances in molecular biology techniques, the genus *Pythium s.l.* was re-examined and divided into 11 phylogenetic clades (from A to K), based on the analysis of the ITS region of rDNA [16]. From this early study, it was clear that this genus was paraphyletic. Subsequently, multiple gene genealogy confirmed this assumption and *Pythium s.l.* was split into five distinct genera. These genera include *Pythium sensu stricto* (hereafter referred to as *Pythium*, encompassing clades A, B, C, and D), *Elangisporangium* (corresponding to clade H), *Globisporangium* (encompassing clades E, F, G, and I), *Phytopythium* (syn. *Ovatisporangium*, corresponding to clade K), and *Pilasporangium*, the last not coinciding with any of the 11 phylogenetic clades [17,18,19]. Each genus has its own unique morphological features, i.e., *Pythium s.s.* produces filamentous, filamentous inflated, or lobate sporangia. *Globisporangium* species produce globose to subglobose sporangia, occasionally with internal proliferation [1,17,18]. In addition, *Phytopythium* species produce ovoid sporangia with internal or external proliferation, resembling sporangia of *Phytophthora* species, while *Elangisporangium* and *Pilasporangium* produce elongated sporangia and sporangia without proliferation, respectively [17,19].

Before the advent of molecular techniques, the identification of species of *Pythium s.l.* was problematic mainly due to pleomorphism of the sexual and asexual structures, the intraspecific phenotypic variability, the inconsistency of isolates when forming some of these structures *in vitro*, and the lack of a comprehensive, sound taxonomic framework [20,21,22,23]. Although molecular techniques along with phylogenetic analyses have substantially assisted in the identification of *Pythium s.l.* species, morphological traits maintain a fundamental taxonomic relevance. Moreover, it is generally recognized that more than one molecular marker is needed for distinguishing most genera and species of oomycetes [23,24,25,26,27,28,29,30].

Several species of *Pythium s.l.* have been reported as rice seedling pathogens [31,32,33,34,35,36,37,38,39,40,41,42,43,44]. However, the diversity of *Pythium s.l.* populations in rice paddies has been little investigated worldwide.

More than 60 diverse taxa of *Pythium s.l.* have been reported from Iran [15,44]. In recent years, various cereal fields in Fars Province of Iran have been surveyed to isolate and identify *Pythium s.l.* species [42,43,44]. These studies revealed that rice paddies are a favorable ecological niche for *Pythium s.l.* species. During the surveys of rice paddies, we recovered two groups of *Pythium s.l.* isolates with distinctive characters that could not be assigned to any known species. Multi-locus phylogenetic inference indicated they were two new clearly distinct taxa. In the present study, these two groups were characterized and formally described as new species.

## 2. Materials and Methods

### 2.1. Isolation

From 2013 to 2015, rhizosphere soil, pond waters, and rice seedlings were randomly sampled in more than 40 diverse rice paddies and nurseries of Fars Province, Iran, most of which showed symptoms of pre- and post-emergence damping off and crown and root rot of seedlings. Geographic coordinates were recorded for each field by Global Positioning System (GPS) (Table 1). Samples were transported to the Mycology Laboratory of the Department of Plant Protection, Shiraz University, for isolation. Roots and basal stems of rice seedlings were washed with distilled water, blotted dry, cut into small segments (2 to 3 mm), and placed on the semi-selective medium for oomycetes CMA-PARP (extract of 40 g/L boiled ground corn; agar 15 g/L; amended with 10 µg/mL pimaricin, 200 µg/mL ampicillin, 10 µg/mL rifampicin, and 25 µg/mL PCNB) [45]. One hundred grams of each soil sample were placed in a plastic container and flooded with tap water to 1 cm above the soil surface [46]. Isolates were recovered from either soil or water samples by baiting with 5 mm surface-sterilized leaf disks of bitter orange (*Citrus aurantium* L.) or 5 mm pieces of sterile meadow grass (*Poa annua* L.) at 25 °C every 8 h for 48 h in total, and plating on CMA-PARP. Isolates were purified by the hyphal tip method on water agar (WA, agar 10 g/L) and stored on CMA (extract of 40 g/L boiled ground corn; agar 15 g/L) slopes at 15 °C.

### 2.2. Morphological Characterization

In order to observe asexual reproductive structures (sporangia, vesicles, and zoospores), isolates were transferred onto CMA containing sterile hemp (*Cannabis sativa* L.) seeds or turfgrass (*Poa* sp.) [47] for 24 h. The hemp seeds or turfgrass were then transferred to Petri dishes containing distilled water [48], sterile soil extract [49], or Schmitthenner solution [50] under a fluorescent light for 48 h and were checked every 8 h six times. Moreover, sporangia formation was examined using French bean agar media (FBA, extract of 30 g/L boiled French bean; agar 15 g/L [) [45] and sterile soil extract. Sexual reproductive structures were obtained with hemp seed agar (HSA, extract of 60 g/L boiled ground hemp seeds; agar 15 g/L) and carrot agar (CA, extract of 250 g/L boiled carrots; agar 15 g/L) incubated in darkness [47]. In order to study colony morphology, isolates were grown on CMA, HSA, CA, potato-dextrose agar (PDA, extract of 300 g/L boiled potatoes; dextrose 20 g/L; agar 15 g/L), and malt extract agar (MEA, malt extract 25 g/L; agar 15 g/L) [47]. Mycelium plugs (5 mm in diameter) from the edge of a 3 d old culture were placed in Petri dishes each containing 20 mm of medium. The dishes were incubated at 25 °C for 48 h. The effect of temperature on mycelium growth rate was tested on potato-dextrose carrot agar (PCA, extract of 20 g/L boiled potatoes; extract of 20 g/L boiled carrots, agar 15 g/L) [2] with three replicate Petri dishes per isolate and per each tested temperature. Dishes were incubated at 0, 5, 10, 15, 20, 25, 30, 35, and 40 °C.

### 2.3. DNA Extraction, PCR, Sequencing, and Phylogenetic Analyses

DNA was extracted using the method described by Mirsoleimani and Mostowfizadeh-Ghalamfarsa (2013) [51]. Primers used for the amplification and sequencing of nuclear (internal transcribed spacers 1, 2, and 5.8S gene of rDNA = ITS; β-tubulin gene = *Btub*) as well as mitochondrial (cytochrome c oxidase subunit II = *cox*2) loci in addition to the PCR conditions of loci are reported in Appendix A. PCR products were purified and sequenced with the primers used for amplification by a dye terminator cycle (Bioneer, Daejeon, South Korea). Sequences were deposited into GenBank. For low-quality ITS sequences, cloning was performed using the Strata Cloning Kit (Agilent Technologies, Santa Clara, CA, USA) according to the manufacturer’s instruction [52].

The resulting sequences were edited and aligned by Geneious Prime 2022 [53] with subsequent visual adjustments. BLAST similarity searches were performed with blastn (for nucleotide-versus-nucleotide comparison) (Appendix A) [54]. Partition homogeneity tests were conducted on combined nuclear and mitochondrial gene alignments by PAUP* 4.0a136 [55] using 100 replicates and a heuristic general search option. To reconstruct the phylogenetic trees, Bayesian inference analyses on individual and concatenated ITS, *Btub*, *cox*1, and *cox*2 loci were carried out with MrBayes 3.1 [56], as implemented in TrEase [57] running 10 M generations with the GTR Gamma + I substitution model and discarding 25% of the initial trees as burnin. In addition, Maximum Likelihood inference was performed using RAxML as implemented in TrEase. All parameters were set to default. The robustness of the Maximum Likelihood trees was estimated by 1000 bootstraps. Phylogenetic trees were edited and displayed with Mega 11 [58].

### 2.4. Pathogenicity

With the exception of the Persepolis region (Table 1), pre- and post-emergence damping off, root rot, and in rare cases, crown rot were observed in 95% of the plants sampled from the selected regions. Hence, the ability of isolates to cause seed rot, stunting, and pre- and post-emergence damping off of rice seedlings was tested in pathogenicity assays. The inoculum was prepared according to the method described by Banihashemi (1989) [59], and Salmaninezhad and Mostowfizadeh-Ghalamfarsa (2019a) [42] using vermiculite amended with 120 mL/L hemp seed extract (extract of 60 g boiled hemp seeds) colonized by the mycelium.

For the pre-emergence damping off tests, rice seeds were washed and planted in pots containing sandy loam soil (500 mL) infested with 10 mL of the inoculum. As a control, rice seeds were planted in pots containing sandy loam soil (500 mL) mixed with 10 mL of sterile vermiculite amended with hemp seed extract. For the post-emergence damping off tests, 20 day old seedlings were transplanted into pots containing sandy loam soil (500 mL) infested with 10 mL of the inoculum. As a control, seedlings were transplanted into pots containing sandy loam soil (500 mL) mixed with 10 mL of sterile vermiculite amended with hemp seed extract. Symptoms were scored two weeks later. Re-isolations were performed from both symptomatic and control seedlings using CMA-PARP medium, according to the method described by [60].

## 3. Results

### 3.1. Morphology of Isolates of Pythium s.l.

Overall, 1169 isolates of *Pythium s.l.* were recovered from rice paddies of Fars Province during the survey. Among them, two groups of isolates with distinctive morphological characteristics were selected for further characterization in this study. Isolates of the first group (16 isolates) produced filamentous to slightly inflated sporangia which released zoospores in aqueous medium. Isolates of the other group (five isolates) produced globose to subglobose hyphal swellings and were not able to produce zoospores. Neither of these two groups of isolates corresponded to the already described species, according to the identification keys of Van der Plaäts-Niterink [2] and Dick [61]. Based on the morphologies of both colonies on different agar media and sexual structures, it was possible to distinguish two different subgroups within the first and more numerous group of isolates, which were designated as Morphotype I and II, respectively (see Section 3.4).

### 3.2. Phylogenetic Analyses

Isolates within each group had identical sequences of nuclear and mitochondrial loci, except for ITS sequences in which some strains showed minor differences. The final alignment length was 877 bp for ITS, 393 bp for *cox*1, 488 bp for *cox*2, 459 bp for *Btub*, and 2045 bp for combined gene regions as regards *Pythium* sp.; and 1317 bp for ITS, 393 bp for *cox*1, 484 bp for *cox*2, 480 bp for *Btub*, and 2709 bp for combined gene regions as regards *Globisporangium* sp. The ITS sequences of isolates of the first group showed 99% similarity with *P. plurisporium* Abad, Shew & L. T. Lucas and two other undescribed *Pythium* species. The ITS sequences of isolates of the second group showed 87 to 88% similarity with *G. coniferarum* Salmaninezhad & Mostowf., three undescribed *Pythium* species, and *G. nagaii* (Ito & Tokun) Uzuhashi, Tojo, & Kakish. The *cox*1 sequences of isolates of the first group showed 98.47% similarity with *P. plurisporium*. The *cox*1 sequences of isolates of the second group showed 97.96% similarity with *G. coniferarum*. The *cox*2 sequences of isolates of the first group showed 96.51% similarity with *P. plurisporium*. The *cox*2 sequences of isolates of the second group showed 98.42% similarity with *G. coniferarum*. The *βtub* sequences of isolates of the first group showed 98.47% similarity with *P. plurisporium*. The *βtub* sequences of isolates of the second group showed 97.41% similarity with *G. coniferarum*. In all phylogenetic trees (Figure 1, Figure 2, Appendix A), each group of isolates formed a well-supported monophyletic group, confirming the conclusion that the two groups were novel species. Bayesian posterior probability was 1.00 for each new lineage in the combined tree and ranged from 0.93 to 1.00 across nuclear and mitochondrial gene trees (Appendix A). Nucleotide differences among the isolates with their sister taxa are shown in Appendix A. The two novel species were designated as *Pythium banihashemianum sp. nov.* and *Globisporangium izadpanahii sp. nov.*, respectively. *Pythium banihashemianum sp. nov.* was located in clade B of the ITS phylogenetic tree and was related to *P. plurisporium*, *P. kashmirense* Paul, *P. afertile* Kanouse & Humphery, *P. rhizo-oryzae* Paul, *P. graminicola* Subraman, *P. vanterpoolii* Kouyeas & Kouyeas, and *P. torulosum* Cocker & Patt.
Figure 1Phylogenetic relationships of *Pythium banihashemianum sp. nov.* from paddy fields of Fars Province among 46 *Pythium sensu stricto* species based on the analysis of multigene genealogies of nuclear (ITS and *Btub*) and mitochondrial (*cox*1 and *cox*2) sequences in Maximum Likelihood tree. Numbers on branches represent posterior probability based on Bayesian analysis and the bootstrap support based on Maximum Likelihood, respectively.
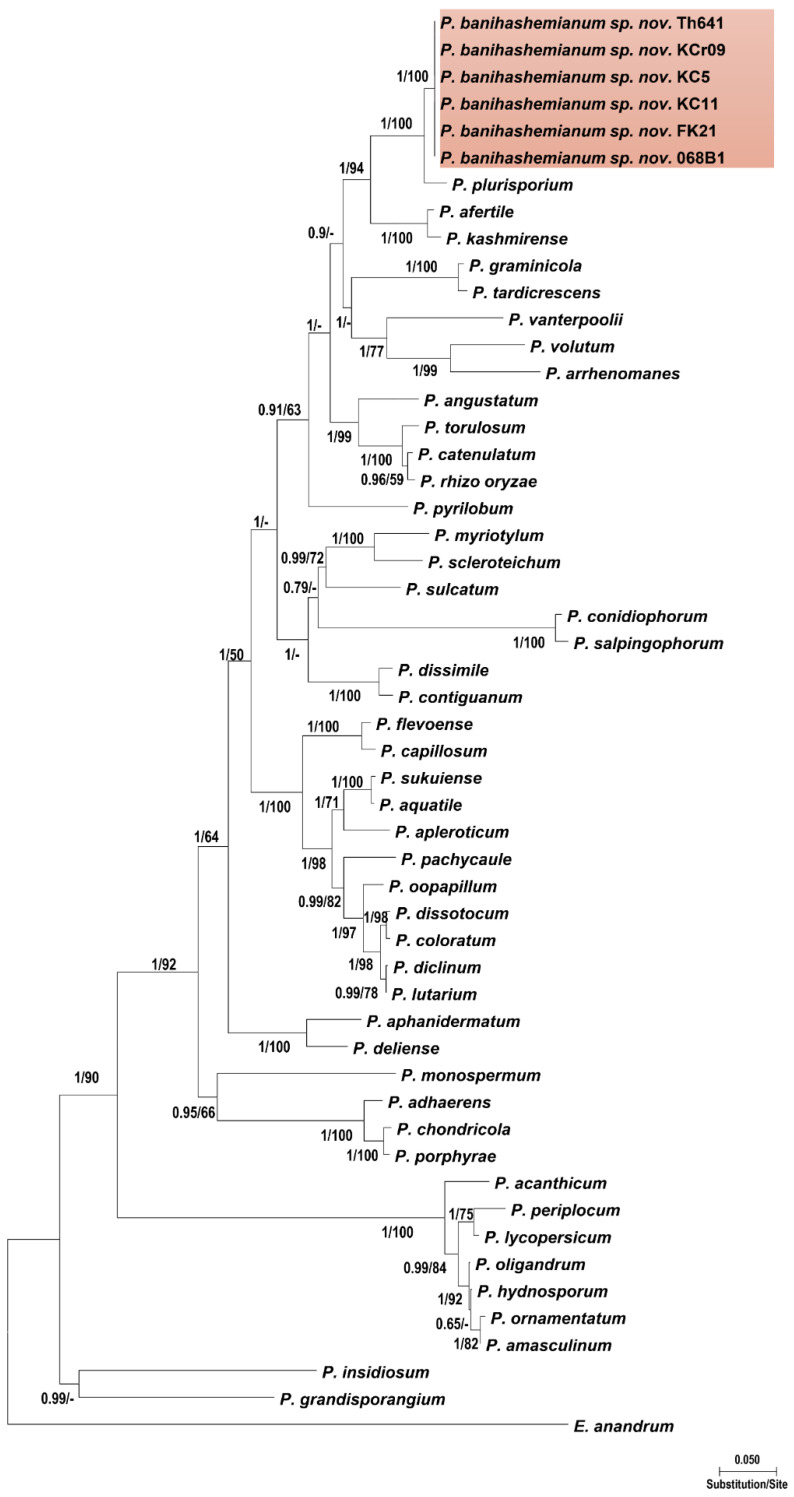

Figure 2Phylogenetic relationships of *Globisporangium izadpanahii sp. nov.* from paddy fields of Fars Province among 46 *Globisporangium* species based on the analysis of multigene genealogies of nuclear (ITS and *Btub*) and mitochondrial (*cox*1 and *cox*2) sequences in Maximum Likelihood tree. Numbers on branches represent posterior probability based on Bayesian analysis and the bootstrap support based on Maximum Likelihood, respectively.
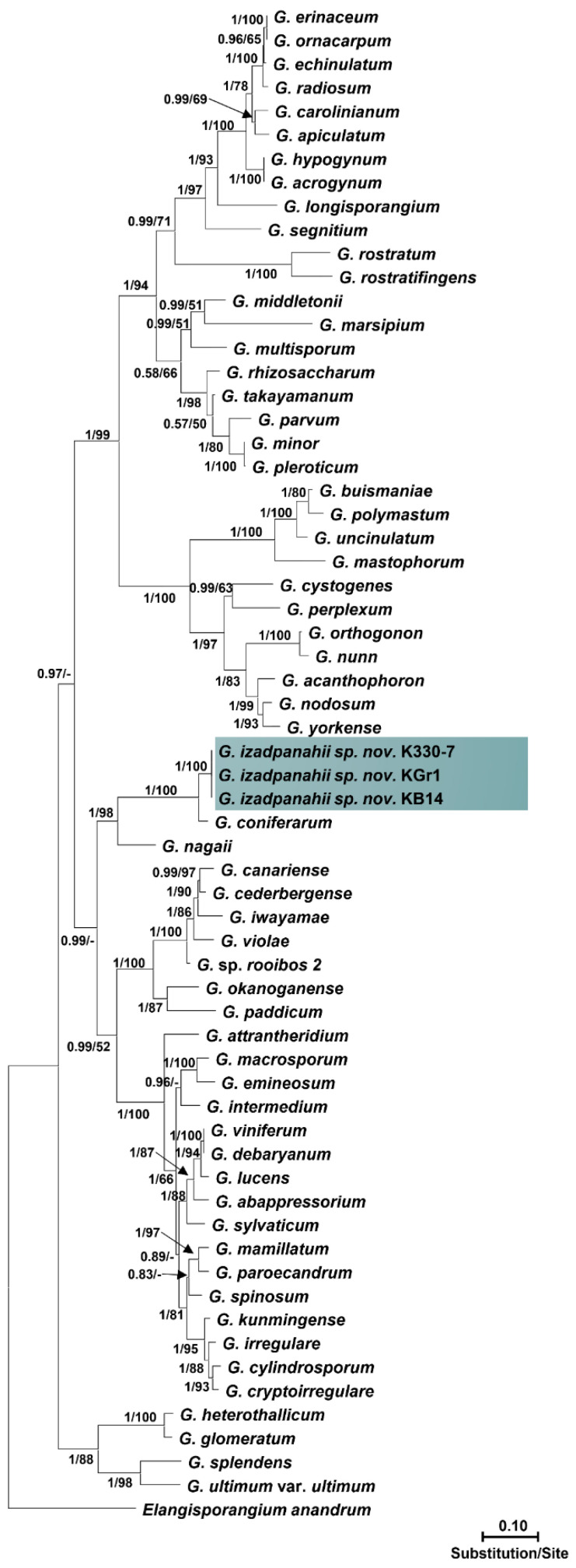



*Globisporangium izadpanahii sp. nov.* was located in clade G of the ITS phylogenetic tree and was related to *G. coniferarum*, *G. nagaii*, *G. okanoganense* (Lipps) Uzuhashi, Tojo, & Kakish, *G. paddicum* (Hirane) Uzuhashi, Tojo, & Kakish, *G. iwayamai* (Ito) Uzuhashi, Tojo, & Kakish, *G. canariense* (Paul) Uzuhashi, Tojo, & Kakish, *G. violae* (Chesters & Hickman) Uzuhashi, Tojo, & Kakish, and *G. cederbergense* (Bahramisharif, Botha, & Lamprecht) Nguyen & Spies. The position of each new species was consistent in all phylogenetic trees. Only the *cox*1 phylogenetic tree of *Globisporangium* species showed polytomy (Appendix A), while no polytomy was observed in other phylogenetic trees (Appendix A).

### 3.3. Pathogenicity

In pathogenicity tests (Table 2), *P. banihashemianum sp. nov.* was pathogenic on rice. The isolates of these species caused pre- and post-emergence damping off, crown rot (Figure 3), seed rot, and a severe decrease in growth rate. They were re-isolated from symptomatic seedlings. Conversely, control seedlings did not show any symptoms. Isolates of *G. izadpanahii sp. nov.* did not induce any disease in seeds or rice seedlings and could not be re-isolated from the roots and crowns of tested plants (Table 2).
Figure 3Pathogenicity tests on roots and crown of rice (*Oryzae sativa*). *Pythium banihashemianum sp. nov.* (068B1) causes root and crown (blue arrow) rot ((**left**) control; (**right**) infected crown and roots). Bar = 1 cm.
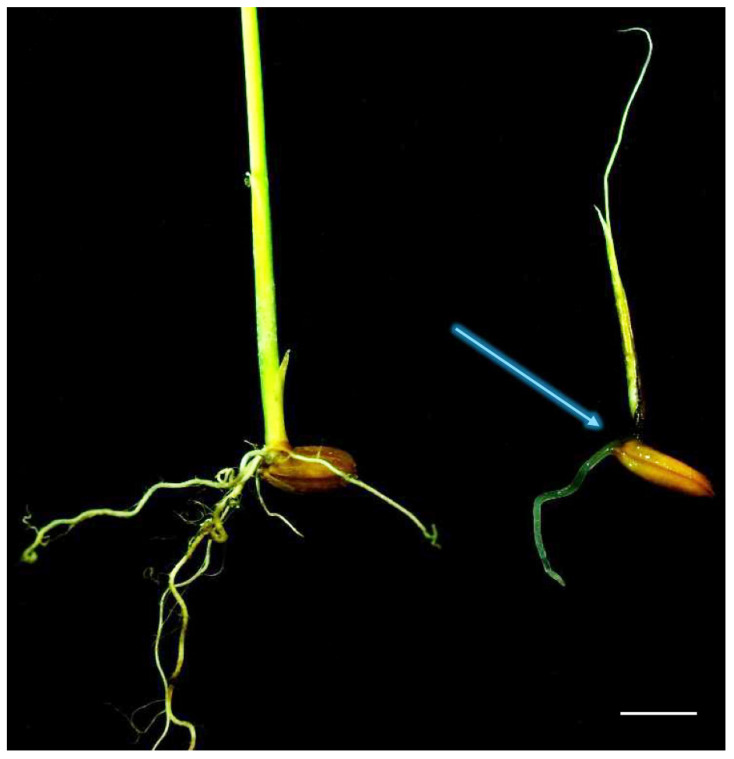



### 3.4. Taxonomy

***Pythium banihashemianum*** Mostowf. & Salmanin. ***sp. nov.*** (Figure 1, Figure 4, Figure 5 and Figure 6)

MycoBank: MB824523

*Typification*: IRAN, Fars Province: Kamfiruz (30°16.934′ N–052°19.155′ E), from roots of *Oryza sativa*, 16 August 2015, F. Salmaninezhad 068B1 (holotype CBS 143876, living culture preserved in a metabolically inactive state at Westerdijk Fungal Biodiversity Institute). GenBank: ITS = KX228083; *βtub* = KX228113; *cox*1 = OP321097; and *cox*2 = KX228120.

*Etymology*: After Prof. Ziaeddin Banihashemi, who is a pioneer in oomycete studies in Iran.

Although no differences were observed in the sequences of isolates assigned to this species, their morphological characteristics were distinct, leading to their division into two morphological groups (morphotypes).

Morphotype I: Colonies on PDA and HSA show a rosette pattern, on CA show an intermediate pattern, and on MEA and CMA show chrysanthemum and radial patterns, respectively (Figure 4a). *Main hypha*: 3.1–4.5 (av. 3.5) μm width. *Sporangia*: not observed on solid media but produce abundantly in aqueous medium containing sterile hemp seeds, filamentous, slightly inflated to rarely dendroid (Figure 5a). *Zoospores*: released through a discharge tube 50–110 μm long. *Hyphal swellings*: not present. *Oogonia*: smooth, rarely globose [29.8–37.4 (av. 33.3) µm], ovoid, jug shaped, sometimes without any specific shape (Figure 5i,j), mostly (more than 80%) with two adjacent projections. *Oogonial projections*: 0.5–0.9 (av. 0.7) µm long (Figure 5c). *Antheridia*: 4–8 per oogonium, clavate, and crook-necked, making apical or lateral contact, paragynous, monoclinous, and diclinous with a very long stalk, which mostly encircles around oogonia (Figure 5j). Each oogonium contains more than one oospore (up to 3). *Oospores*: aplerotic, globose to subglobose, 28.1–35.5 (av. 32.5) µm diam., with a wall 1.4–2 (av. 1.7) µm thick. Oospore formation is specific, and the oogonium stalk initially swells, leading to the first oospore formation. Subsequently, the terminal section of the oogonium swells, and the oospore moves into this section resulting in the formation of a second oospore in the oogonium swollen stalk (Figure 5c). Comparisons of the morphological characteristics of the isolates assigned to *P. banihashemianum sp. nov.* Morphotype I with its sister taxa are shown in Table 3. Morphometric features of the examined isolates are shown in Appendix A. Colonies on PCA have an average radial growth rate of 2.5 mm d^−1^ at 10 °C, 5 mm d^−1^ at 15 °C, 7 mm d^−1^ at 20 °C, 10 mm d^−1^ at 25 °C and 30 °C, 11 mm d^−1^ at 35 °C, 1 mm d^−1^ at 40 °C, and no growth occurred at 5 °C. *Cardinal temperatures*: minimum 10 °C, optimum 35 °C, and maximum 40 °C (Figure 6).
Figure 4Colony morphology of *Pythium banihashemianum sp. nov.* Morphotype I isolate 068B1 (**a**), *P. banihashemianum sp. nov.* Morphotype II, isolate KC5 (**b**) after 24 h on various media at 25 °C; **top** (from left to right): carrot agar, malt extract agar, and potato-dextrose agar; **bottom** (from left to right): cornmeal agar and hemp seed agar. Bar = 1 cm.
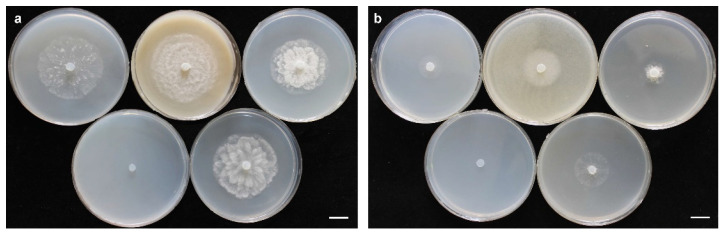

Figure 5Morphological structures of *Pythium banihashemianum sp. nov*. (**a**) Filamentous sporangium (Morphotype I); (**b**) catenulate oospores with two paragynous antheridia (arrows) per oogonium (Morphotype II); (**c**) formation of oospore in an oogonium with two papillae (Morphotype I); (**d**) oogonium with single oospore and a long papilla (arrows; Morphotype II); (**e**) slightly inflated filamentous sporangium; (**f**) oogonium with single oospore showing two symmetrical papillae (Morphotype II); (**g**) catenulate oospores with both mono- and diclinous antheridia (Morphotype II); (**h**) catenulated aplerotic oospore with diclinous antheridium (arrows; Morphotype II); (**i**) oogonium with two oospores (**left**) and aplerotic oospore with two antheridia (**right**) (Morphotype I); (**j**) amorphous catenulated oogonia with hypogynous antheridium (arrows; Morphotype I). Bars = 10 µm, except for e and o where Bar = 20 µm.
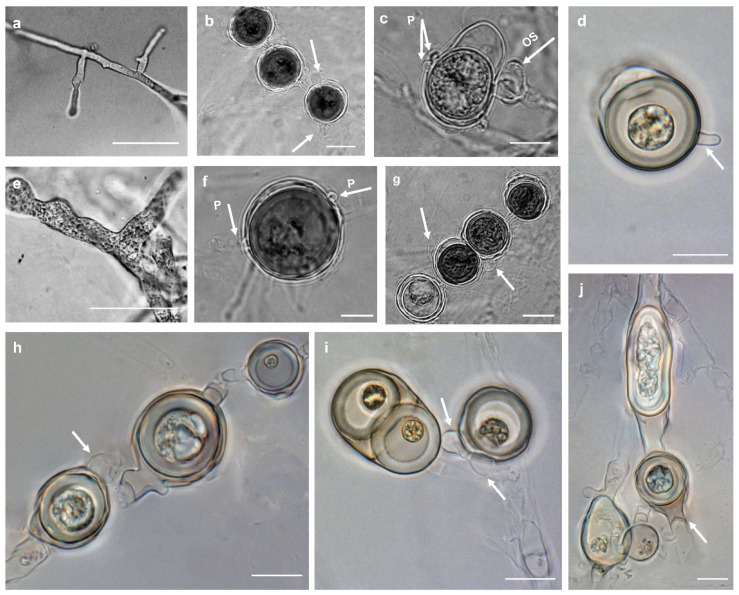

Figure 6Average radial growth rates of *Pythium banihashemianum sp. nov.* (16 isolates) and *Globisporangium izadpanahii sp. nov.* (five isolates) on potato-dextrose carrot agar at different temperatures.
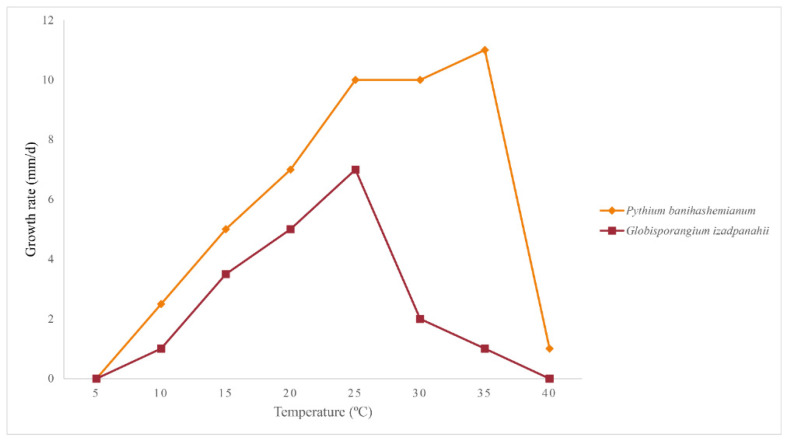



Morphotype II: Colonies show a radial pattern on CA and HSA, a uniform pattern on CMA, an intermediate pattern on PDA, and no specific pattern on MEA (Figure 4b). *Main hyphae*: 2.7–4.0 (av. 3.1) μm in width. *Sporangia*: filamentous and inflated, never observed on solid media and produced abundantly on aqueous medium with sterile hemp seeds. (Figure 5e). *Zoospores*: released after 12 h from 61–115 μm long discharge tubes. The mycelium grows easily on HSA and CA, producing abundant oogonia, antheridia, and oospores. Two kinds of oospore formation are observed: single (Figure 5f) and catenulate (Figure 5b,g). *Oogonia*: smooth, globose 28.1–40.5 (av. 38.3) μm, terminal, mostly with more than one oospore (up to 5 with catenulate formation) (Figure 5b,g,h). More than 80% of the oogonia contain two papillae on both sides which are 1.1–2.3 (av. 1.7) µm long (Figure 5d). *Antheridia*: 1–2 per oogonia with catenulate oospores and rarely (less than 5%) up to 4 per oogonia with a single oospore, crook-necked, making apical contact, paragynous, mostly monoclinous, rarely diclinous. *Oospores*: globose, aplerotic, smooth, most (more than 90%) catenulate, 26.7–36.4 (av. 32.4) µm diam., with a wall 0.8–3.0 (av. 1.5) µm thick. Comparisons of the morphological characteristics of the isolates assigned to *P. banihashemianum sp. nov.* Morphotype II with its sister taxa are shown in Table 3. Morphometric features of the examined isolates are shown in Appendix A. Colonies on PCA have an average radial growth rate of 1 mm d^−1^ at 5 °C, 2 mm d^−1^ at 10 °C, 3 mm d^−1^ at 15 °C, 5 mm d^−1^ at 20 °C, 7 mm d^−1^ at 25 °C, 12 mm d^−1^ at 30 °C, 9 mm d^−1^ at 35 °C, and 1 mm d^−1^ at 40 °C. *Cardinal temperatures*: minimum 5 °C, optimum 30 °C, and maximum 40 °C (Figure 6).

*Other specimens examined*: IRAN. Fars Province: Kamfiruz (30°11.017′ N–052°27.900′ E), from rhizosphere of *Oryzae sativa*, 16 August 2015, F. Salmaninezhad K116-1. IRAN, Fars Province: Kamfiruz (30°11.845′ N–052°27.787′ E), from pond water of paddy fields, 20 May 2014, F. Salmaninezhad K101-4. IRAN, Fars Province: Kamfiruz (30°11.911′ N–052°27.777′ E), from rhizosphere of *O. sativa*, 20 May 2014, F. Salmaninezhad G112-2. IRAN, Fars Province: Kamfiruz (30°11.909′ N–052°27.779′ E), from the soil of paddy fields, 20 May 2014, F. Salmaninezhad 056S2. IRAN, FARS Province: Ramjard (30°07.274′ N–052°32.946′ E), from rhizosphere of *O. sativa*, 9 November 2015, F. Salmaninezhad 033B7. IRAN, Fars Province: Ramjard (30°07.234′ N–052°32.983′ E), from roots of *O. sativa*, 9 November 2015, F. Salmaninezhad 038C3. IRAN, Fars Province: Ramjard (30°02.780′ N–052°49.513′ E), from the roots of *O. sativa*, 9 November 2015, F. Salmaninezhad 048S1. IRAN, Fars Province: Firuz Abad (28°51.587′ N–052°30.842′ E), from rhizosphere of *O. sativa*, 20 May 2014, F. Salmaninezhad F201-3. IRAN, Fars Province: Firuz Abad (28°51.407′ N–052°30.666′ E), from *O. sativa* roots, 9 November 2015, F. Salmaninezhad Fk21. GenBank: ITS = MK454539; *βtub* = MK540655; *cox*1 *=* OP321098; *cox*2 = MK455862. IRAN, Fars Province: Firuz Abad (28°49.735′ N–052°29.149′ E), from *O. sativa* roots, 9 November 2015, F. Salmaninezhad Fs301. IRAN, Fars Province: Firuz Abad (28°49.989′ N–052°29.551′ E), from rhizosphere of *O. sativa*, 20 May 2014, F. Salmaninezhad F32-01. IRAN, Fars Province: Persepolis (29°59.008′ N–052°49.513′ E), from rhizosphere of *O. sativa*, 16 August 2015, F. Salmaninezhad Th641. GenBank: ITS = MK454538; *βtub* = MK540656; *cox*1 *=* OP321102; *cox*2 = MK455863. IRAN, Fars Province: Persepolis (29°58.892′ N–052°57.734′ E), from rhizosphere of *O. sativa*, 16 August 2014, F. Salmaninezhad KC5. GenBank: ITS = MK454707; *βtub* = MK455865; *cox*1 = OP321099; *cox*2 = MK455856. IRAN, Fars Province: Ramjard (30°05.901′ N–052°35.482′ E), from *O. sativa* roots, 16 August 2014, F. Salmaninezhad KCr09. GenBank: ITS = MK454706; *βtub* = MK455864; *cox*1 = OP321101; *cox*2 = MK455857. IRAN, Fars Province: Ramjard (30°05.476′ N–052°35.563′ E), from *O. sativa* crown, 9 November 2015, F. Salmaninezhad KC11 (CBS 143875). MB824524. GenBank: ITS = KX228081; *βtub* = MK455866; *cox*1 *=* OP321100; *cox*2 = MK455858.

*Notes*: This species belongs to clade B of the ITS phylogenetic tree *sensu* Lévesque and de Cock [16] and is closely related to *P. plurisporium* (Figure 1). *Pythium banihashemianum sp. nov.* differs from all other *Pythium* species from clade B due to its high-temperature tolerance and amorphous oogonia with more than one oospore, and from *P. plurisporium* due to it producing a high proportion of papillate oogonia, containing more than one papilla in most oogonia, the presence of two papillae on oogonia, the special formation of oospores, and its unique sequences of mitochondrial and nuclear genes. Adjacent papillae were abundant in isolate 068B1.

***Globisporangium izadpanahii*** Salmanin. & Mostowf. ***sp. nov.*** (Figure 2, Figure 4, Figure 6, Figure 7 and Figure 8)

MycoBank: MB824525

*Typification*: IRAN, Fars Province: Firuz Abad (28°49.989′ N–052°29.551′ E), from rhizosphere of *Oryzae sativa* nursery, 9 November 2015, F. Salmaninezhad K330-7 (holotype CBS 144006, living culture preserved in a metabolically inactive state at Westerdijk Fungal Biodiversity Institute). GenBank: ITS = MK454537; *βtub* = MK455869; *cox*1 = OP321103, *cox*2 = MK455859.

*Etymology*: After Prof. Keramatollah Izadpanah, who is a leading phytopathologist in Iran.

Colonies on PDA and MEA show a rosette pattern, and on HSA, CMA, and CA, a radial pattern (Figure 7). *Sporangia and zoospores*: not produced. *Hyphal swelling*: terminal or intercalary, formed in aqueous medium after one week, 12.9–13.7 (av. 13.2) µm in diam, never observed on solid media (Figure 8a). *Main hyphae*: 4.0–4.8 (av. 4.3) µm in width. *Oogonia*: globose, smooth, terminal, or intercalary, 62.0–63.9 (av. 63.0) µm diam (Figure 8c–d), and most contain a needle-shaped papilla up to 0.8–3.1 (av. 1.0 µm) long (Figure 8d). *Antheridia*: just one per oogonium, crook-necked, elongated, and clavate, mostly monoclinous, rarely diclinous, making apical contact with the oogonium, paragynous and sometimes hypogynous (Figure 8b,d). *Oospores*: globose, perfectly plerotic, with a wall which is up to av. 9.2 µm thick. Comparisons of the morphological characteristics of the isolates assigned to *G. izadpanahii sp. nov.* with its sister taxa are shown in Table 3. Morphometric features of the examined isolates are shown in Appendix A. Colonies on PCA have an average radial growth rate of 1 mm d^−1^ at 10 °C, 3.5 mm d^−1^ at 15 °C, 5 mm d^−1^ at 20 °C, 7 mm d^−1^ at 25 °C, 2 mm d^−1^ at 30 °C, 1 mm d^−1^ at 35 °C, and no growth at 5 °C and 40 °C. *Cardinal temperatures*: minimum 10 °C, optimum 25 °C, and 35 °C (Figure 6).

*Other specimens examined*: IRAN, Fars Province: Kamfiruz (29°58.823′ N–052°53.651′ E), from *Oryzae sativa* crown, 9 November 2015, F. Salmaninezhad KGr1. GenBank: ITS = MK454535; *βtub* = MK455867; *cox*1 = OP321105; *cox*2 = MK455861. IRAN, Fars Province: Ramjard (30°06.139′ N–052°26.892′ E), from rhizosphere of *Oryzae sativa*, 9 November 2015, F. Salmaninezhad Rfa01. IRAN, Fars Province: Kamfiruz (30°18.134′ N–052°17.767′ E), from rhizosphere of *O. Sativa*, 9 November 2015, F. Salmaninezhad Kha3. IRAN, Fars Province: Kamfiruz (30°19.236′ N–052°16.560′ E), from pond water of paddy fields, 9 November 2015, F. Salmaninezhad KB14. GenBank: ITS = MK454536; *βtub* = MK455868; *cox*1 = OP321104; *cox*2 = MK455860.

*Notes*: This species belongs to clade G of the ITS phylogenetic tree *sensu* Lévesque and de Cock [16] in the vicinity of *G. coniferarum* and *G. nagaii* (Figure 1)*. Globisporangium izadpanahii sp. nov.* does not form sporangia and zoospores under standard conditions tested including different temperatures. However, the formation of hyphal swellings in aqueous medium after one week, the unique type of oogonia with a long needle-shaped papilla, its strictly plerotic oospores, it special and unique growth pattern on various media, and, especially, the presence of an elongated clavate antheridium differentiated this species from other known *Globisporangium* species. Additionally, the unique sequences of mitochondrial and nuclear genes separated *G. izadpanahii sp. nov.* from other species.
Figure 7Colony morphology of *Globispirangium izadpanahii sp. nov.* isolate K330-7 after 24 h on various media at 25 °C; (**top**) (from left to right): carrot agar, malt extract agar, and potato-dextrose agar; (**bottom**) (from left to right): cornmeal agar and hemp seed agar. Bar = 1 cm.
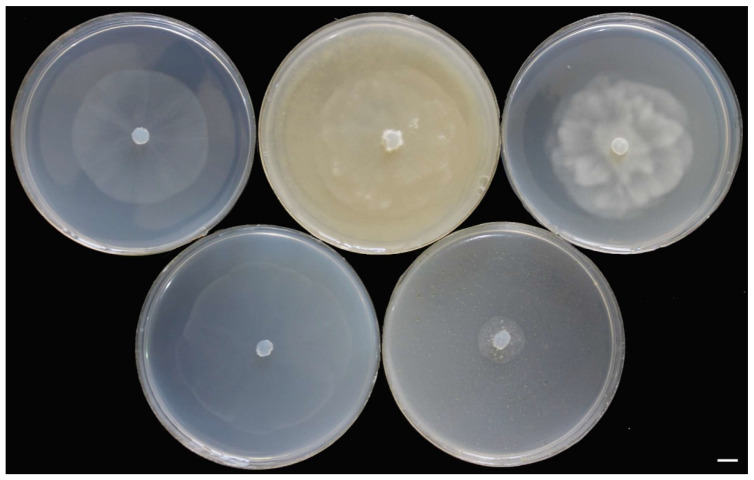

Figure 8Morphological structures of *Globisporangium izadpanahii sp. nov*. (**a**) hyphal swellings; (**b**) intercalary oogonium with a single clubbed shape antheridium; (**c**) smooth oogonium with paragynous antheridium; (**d**) perfectly plerotic oospore with a long papilla. Bars: = 10 µm.
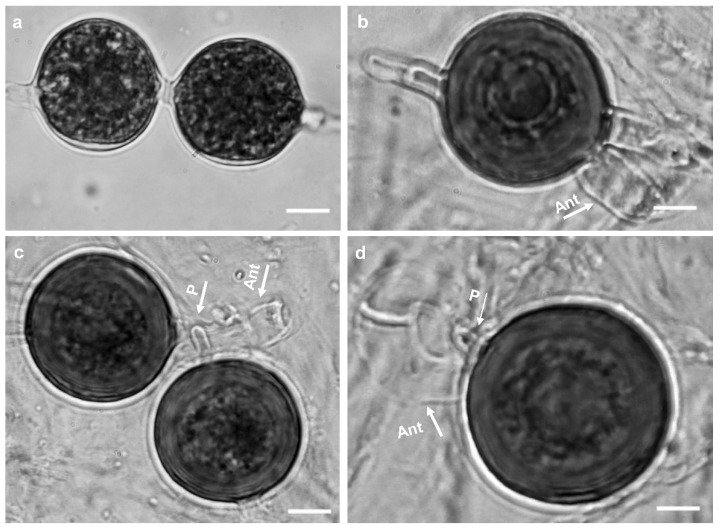



## 4. Discussion

This study is part of a larger research project aimed at investigating the diversity of *Pythium s.l.* populations in rice paddies of Fars Province in Iran. Among more than a thousand *Pythium s.l.* isolates recovered, 16 already known species and three new *Pythium* species, *P. heteroogonium*, *P. longipapillum*, and *P. oryzicollum*, had previously been identified on the basis of morphological and molecular traits [42,44]. In the present study, two groups of isolates from the same large set of isolates, previously recovered from rice paddies, were characterized. The two groups, showing distinctive morphological characters and forming two separate well-supported monophyletic lineages, were formally described as new species, *P. banihashemianum sp. nov.* and *G. izadpanahii sp. nov.*, respectively. The species diversity of *Pythium s.l.* in rice paddies of Fars Province [42,44] indicate that the aquatic environment of this peculiar type of managed ecosystem provides a favorable ecological niche to these oomycetes.

*Pythium banihashemianum sp. nov.* was grouped within clade B of the ITS phylogenetic tree of *Pythium sensu stricto*, but in a separate lineage from other known species. The closest relatives of this species are *P. plurisporium*, *P. kashmirense*, and *P. afertile.* The isolates assigned to *P. banihashemianum sp. nov.* were in turn split into two diverse morphotypes. Both morphotypes produced filamentous-type sporangia. However, the sporangia produced by Morphotype I were mostly dendroid while Morphotype II produced mostly inflated sporangia. In contrast to Morphotype I, which produced one to three oospores in a single oogonium, Morphotype II produced mostly one oospore per oogonium. In addition, the presence of asymmetrical oogonia with papillae separates Morphotype I from Morphotype II, which formed catenulated globous oogonia. The number of antheridia per oogonium was up to eight in Morphotype I and occasionally up to four in Morphotype II. Thus, great intraspecific morphological variability is a rare yet interesting phenomenon that was previously reported for *P. plurisporium*, a closely related species to *P. banihashemianum sp. nov.*, [62]. *Pythium plurisporium* isolates form two different types of oogonia, with a single oospore or with more than one oospore [62,63]. While both *P. banihashemianum sp. nov.* and *P. plurisporium* may produce more than one oospore per oogonium, the maximum number of oospores per oogonium in *P. banihashemianum sp. nov.* (up to at most 3) is less than that in *P. plurisporium* (up to 6) [62,63]. Moreover, the ability of *P. banihashemianum sp. nov.* to produce single oospores clearly separates it from *P. plurisporium* and any other known *Pythium* species.

The existence of sexual structures in *P. banihashemianum sp. nov.* clearly separates it from *P. afertile*, which does not reproduce sexually. *Pythium kashmirense* and Morphotype II of *P. banihashemianum sp. nov.* have sporangial type in common; however, in *P. kashmirense*, antheridial filaments coil around the oogonial stalks [64]. Although ITS and *βtub* phylogenetic trees could not differentiate *P. banihashemianum sp. nov.* from *P. plurisporium*, mitochondrial loci phylogeny clearly separated them from each other, indicating *P. banihashemianum sp. nov.* is a new distinct species. Consistently with Robideau et al. [23] and Hyde et al. [1], more than one gene phylogeny was needed to separate *P. banihashemianum sp. nov.* from *P. plurisporium*.

Although polytomy was observed in *cox*1 loci analyses, *G. izadpanahii sp. nov.*’s location in clade G was highly supported by analyzing other loci. Polytomy in *cox*1 analyses was also observed previously in *Pythium s.l.* [1,23]. A specific type of antheridia, and the absence of sporangia, vesicles, or zoospores separated *G. izadpanahii sp. nov.* from other described species. *Globisporangium izadpanahii sp. nov.* exhibits significant variations in form and structure when compared to its sister species *G. coniferarum* and *G. nagaii*. In contrast to *G. izadpanahi sp. nov.*, *G. coniferarum* produces ovoid to ellipsoid sporangia with vesicles and zoospores [54]. Furthermore, one of the key characteristics of *G. coniferarum* is the production of abundant chlamydospores and different shapes of oogonia (from globose to ovoid and ellipsoid) [52], while in *G. izadpanahii sp. nov.* chlamydospores and oogonia were exclusively globose. The production of terminal, ovoid to pyriform, and proliferating sporangia in *G. nagaii* [2] also separates it from *G. izadpanahii sp. nov.* Additionally, in *G. nagaii*, oogonia are terminal and globose, with aplerotic oospores, and the antheridia disappear soon after fertilization [2]. Such characters have never been observed in *G. izadpanahii sp. nov.* Even though *G. izadpanahii sp. nov.* is a member of clade G, other members of this clade show no similar morphological characteristics to this species.

The presence of multiple divergent copies of the ITS region is a well-known phenomenon among *Pythium s.l.* species and was reported previously for *G. coniferarum* [52]. Analysis of the ITS sequences of *Globisporangium izadpanahii sp. nov.* clones showed that the ITS region had many insertions, which led to the overlapping of the direct sequences, disruption of the electropherograms, and, consequently, low-quality sequences. Furthermore, *G. izadpanhaii sp. nov.* isolates showed intraspecific ITS sequence heterogeneity. Using the resulting contigs of ITS clones, we showed that *G. izadpanahii sp. nov.* is a new species located in a separate lineage close to *G. nagaii.* Other loci sequences (i.e., *Btub*, *cox*1, and *cox*2) showed a very high quality and strongly supported the separation of this new species.

Although all isolates characterized in this study were recovered from rice paddies and rice plants showing severe symptoms of root and crown rot, pathogenicity tests revealed that *P. banihashemianum sp. nov.* isolates were extremely pathogenic and caused severe symptoms on rice seedlings, while *G. izadpanahii sp. nov.* isolates were not pathogenic on rice seedlings.

The discovery of two new species, *P. banihashemianum sp. nov.* and *G. izadpanahii sp. nov.*, provides new insight into the diversity of oomycetes. The advent of molecular techniques and multigene phylogenetic analysis has greatly contributed to the advancement of the systematics of oomycetes, and they are valuable tools when setting up a sound internationally recognized taxonomic framework. Numerous new cryptic species were separated from species complexes and polyphyletic genera, like *Pythium s.l.*, and were split into clearly separate genera, differing also in morphological and physiological features [19,65]. However, a putative new taxon cannot be regarded as a species until its hierarchical status has been determined and its name has been formally designated according to the rules of the International Code of Nomenclature for algae, fungi, and plants [28]. The precise identification of a taxon has also phytopathological implications and is crucial for the effective management of plant diseases caused by oomycetes. For instance, of the two new species described in this study, both associated to symptomatic rice seedlings, only *P. banihashemianum sp. nov.* was proved to be pathogenic.

## 5. Conclusions

The description of two novel species, *P. banihashemianum sp. nov.* and *G. izadpanahii sp. nov.*, in addition to those identified in previous studies, contributes to the advancement of the systematics of *Pythium s.l.* and supports its segregation into several distinct genera. Moreover, this study confirms that rice paddies are a wide repository of diversity of these oomycetes, which thrive in this type of ecosystem. In pathogenicity tests on rice seedlings, *P. banihashemianum sp. nov.* was proved to be an aggressive pathogen, while *G. izadpanahii sp. nov.* was not pathogenic, indicating species of *Pythium s.l.* may have multiple ecological roles in rice paddies. The agronomic, phytopathological, and taxonomic relevance of unveiling the diversity of *Pythium s.l.* populations in managed ecosystems, such as rice paddies, will encourage the extension of this study to other geographic areas and diverse cereal crops.

## Figures and Tables

**Table 1 jof-10-00405-t001:** List of *Pythium sensu lato* isolates recovered from rice paddy fields of Fars Province of Iran with their GenBank accession numbers.

Species	Isolates	Date of Collection	Location	Longitude	Latitude	Matrix	GenBank Accession Number			
							ITS ^a^	*Btub* ^b^	*cox*1 ^c^	*cox*2 ^d^
*Pythium banihashemianum sp. nov.*										
	068B1 *^†^	August 2015	Kamfiruz	30°16.934′ N	052°19.155′ E	Rice root	KX228083	KX228113	OP321097	KX228120
	Th641 ^†^	August 2015	Persepolis	29°59.008′ N	052°49.513′ E	Rice soil	MK454538	MK540656	OP321102	MK455863
	Fk21 ^†^	November 2015	Fiurz Abad	28°51.407′ N	052°30.666′ E	Rice root	MK454539	MK540655	OP321098	MK455862
	048S1 ^†^	November 2015	Ramjard	30°02.780′ N	052°49.513′ E	Rice root	N/A ^‡‡^	N/A	N/A	N/A
	038C3 ^†^	November 2015	Ramjard	30°07.234′ N	052°32.983′ E	Rice root	N/A	N/A	N/A	N/A
	033B7 ^†^	November 2015	Ramjard	30°07.274′ N	052°32.946′ E	Rice soil	N/A	N/A	N/A	N/A
	056S2 ^†^	May 2014	Kamfiruz	30°11.909′ N	052°27.779′ E	Rice soil	N/A	N/A	N/A	N/A
	K116-1 ^†^	August 2015	Kamfiruz	30°11.017′ N	052°27.900′ E	Rice soil	N/A	N/A	N/A	N/A
	Fs301 ^†^	November 2015	Fiurz Abad	28°49.735′ N	052°29.149′ E	Rice root	N/A	N/A	N/A	N/A
	F32-01 ^†^	May 2014	Fiurz Abad	28°49.989′ N	052°29.551′ E	Rice soil	N/A	N/A	N/A	N/A
	F201-3 ^†^	May 2014	Fiurz Abad	28°51.587′ N	052°30.842′ E	Rice soil	N/A	N/A	N/A	N/A
	KC11 **	November 2015	Ramjard	30°05.476′ N	052°35.563′ E	Rice crown	KX228081	MK455866	OP321100	MK455858
	KC5 **	August 2014	Persepolis	29°58.892′ N	052°57.734′ E	Rice soil	MK454707	MK455865	OP321099	MK455856
	KCr09 **	August 2014	Ramjard	30°05.901′ N	052°35.482′ E	Rice root	MK454706	MK455864	OP321101	MK455857
	G112-2 **	May 2014	Kamfiruz	30°11.911′ N	052°27.777′ E	Rice soil	N/A	N/A	N/A	N/A
	K101-4 **	May 2014	Kamfiruz	30°11.845′ N	052°27.787′ E	Pond water	N/A	N/A	N/A	N/A
*Globisporangium izadpanahii sp. nov.*										
	K330-7 ^‡^	November 2015	Firuz Abad	28°49.989′ N	052°29.551′ E	Soil	MK454537	MK455869	OP321103	MK455859
	KGr1	November 2015	Kamfiruz	29°58.823′ N	052°53.651′ E	Rice crown	MK454535	MK455867	OP321105	MK455861
	KB14	November 2015	Kamfiruz	30°19.236′ N	052°16.560′ E	Pond water	MK454536	MK455868	OP321104	MK455860
	Rfa01	November 2015	Ramjard	30°06.139′ N	052°26.892′ E	Soil	N/A	N/A	N/A	N/A
	KHa3	November 2015	Kamfiruz	30°18.134′ N	052°17.767′ E	Rice root	N/A	N/A	N/A	N/A

^a^ Internal transcribed spacers 1, 2, and 5.8S gene of rDNA. ^b^ β-tubulin. ^c^ Cytochrome c oxidase subunit I. ^d^ Cytochrome c oxidase subunit II. * = CBS 143876, type species; ^‡^ = CBS 144006, type species, ^†^ = Morphotype I, ** = Morphotype II, ^‡‡^ Not Applicable.

**Table 2 jof-10-00405-t002:** Pathogenicity results of the *Pythium sensu lato* species examined in this study.

Species	Isolate Code	Pathogenicity on Rice	Symptom					
			Post-Emergence Damping Off (%)	Pre-Emergence Damping Off (%)	Seed Rot (%)	Stunting (%)	No Growth (%)	Host Tissue Colonization *
*Pythium banihashemianum sp. nov.*								
	068B1 **	+	80	70	80	0	60	+
	Fk21 **	+	70	60	90	0	70	+
	Th641 **	+	90	60	40	0	50	+
	KC11 ^†^	+	70	90	80	0	80	+
	KC5 ^†^	+	80	60	60	0	70	+
	KCr09 ^†^	+	90	50	70	0	50	+
*Globisporangium izadpanahii sp. nov.*								
	K330-7	−	0	0	0	0	0	−
	KB14	−	0	0	0	0	0	−
	KGr1	−	0	0	0	0	0	−

* (+) positive and (−) negative results, ** = Morphotype I, ^†^ = Morphotype II.

**Table 3 jof-10-00405-t003:** Morphological comparison of the species described in this study with their related species.

Character	*G. izadpanahii*	*G. coniferarum*	*G. nagaii*	*P. banihashemianum* (Morphotype I: MT1, Morphotype II: MT2)	*P. plurisporium*	*P. afertile*	*P. kashmirense*
Cardinal Temperatures (°C)	5; 35; 40	15; 30; 40	9; 28; 35	MT1:10; 35; 40 MT2:5; 30; 40	15; 35; 40	5; 25; 30	5; 25; 35
Daily Growth Rate at 25 °C (mm)	7	6	25	MT1:10 MT2:7	10	18	15
Colony Pattern on CMA	Radial	Radial	No data	MT1; Radial MT2: Uniform	Radial	Compact colony without aerial mycelium	No data
Hyphae (µm)	4.3	3.1–7.3	4	MT1:3.1 MT2:3.5	8	6.6	5–6
Chlamydospore	−	(+), (sub)globose, ovoid	−	−	−	+	−
Sporangium or Hyphal Swelling Production (µm)	(+) 13.2	(+) 16.3–17.6	(+)	(+) Variable	(+) Variable	(+) Variable	(+) Variable
Sporangium or Hyphal Swelling Shape	Globose	Ellipsoid, globose, ovoid, with pedicel	Ovoid, pyriform	MT1: filamentous, inflated; MT2: Filamentous, slightly inflated	Filamentous, inflated	Strictly filamentous	Filamentous, inflated, contiguous
Sporangium or Hyphal Swelling Position	Terminal or intercalary	Mostly terminal	Terminal	−	−	−	−
Sporangium Proliferation	−	−	+	−	−	−	−
Zoospore Production	−	+	+	+	+	+	+
Homothallic/Heterothallic	Homothallic	Homothallic	Homothallic	Homothallic	Homothallic	No sexual structure	Homothallic
Oogonium Ornamentation	Smooth with a papillae	Smooth	Smooth	Smooth with 2 papilla	Smooth	No sexual structure	Smooth
Oogonium Shape and Dimensions (µm)	Globose, 63	Globose, ovoid, ellipsoid, 25.9–26.7	Globose, sometimes irregular, 14–22	MT1: Ovoid, amorphous, 33.3; MT2: Globose, 38.3	Globose to obpyriform, 23–35	−	Globose, 16.38
Antheridium Shape	Crook-necked, elongated, clavate	Clavate to no specific shape	Clavate	MT1 & MT2: Clavate, crook-necked	Crook-necked	−	Coiled around oogonium
Antheridium Type and Number per Oogonium	Mostly monoclinous, 1	Monoclinous and diclinous, paragynous, rarely hypogynous, 1–5	Monoclinous, 1 (disappears after fertilization)	MT1: Monoclinous and diclinous, 4–8 MT2: mostly monoclinous, 1–4	Monoclinous, and Diclinous, 2–12	−	Diclinous1–8
Oospore Type and Dimensions (µm)	Perfectly plerotic, 63	Aplerotic and plerotic 21.3–23.2	Aplerotic, 12–19	MT1: Aplerotic, 32.5 MT2: Aplerotic, 32.4	Aplerotic, 26.2–30	−	Plerotic, 16.11
Oospore Wall (µm)	9.2	1.9–3.1	0.8	MT1: 1.7 MT2:1.5	2.2–3	−	1.27
Number of Oospore per Oogonium	1	1	1	MT1: 1–2 MT2: 1	1–6	−	1

## Data Availability

The datasets generated and analyzed during the current study are in Appendix A or available from the corresponding author on reasonable request.

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
