# Peer review of "Pythium banihashemianum sp. nov. and Globisporangium izadpanahii sp. nov.: Two New Oomycete Species from Rice Paddies in Iran"

_jof, 2024, doi:10.3390/jof10060405_

Round 1

Reviewer 1 Report

  1. Could the authors elaborate on the criteria used for morphological and physiological differentiation of the newly identified species from closely related species?
  2. How were the sampling locations within Fars Province selected, and do these locations comprehensively represent the rice paddy ecosystem in the region?
  1. What methods were employed to ensure the accuracy of the phylogenetic analysis, particularly regarding the selection of genes for Bayesian inference and Maximum Likelihood analyses?
  1. Is there evidence to suggest a broader geographical distribution of these species beyond Fars Province, or are they endemic to this region?
  1. 30% iThenticate Percent match looks too high.
  2. Figures are not clear.
  3. Table S1 is missing. 
  4. English is very difficult to understand.

Author Response

Dear Reviewer, the point-by-point response letter is attached for your convenience.

Kind regards

Santa Olga Cacciola

Reviewer 2 Report

In this paper, the authors describe and study two new species of oomycetes, named Pythium banihashemianum sp. nov. and Globisporangium izadpanahii sp. nov.. Studying the diversity of microorganisms in the environment is an important task, since they can be extremely interesting for agriculture, molecular biology or other areas of science. The authors carried out the ideatification based on morphological and physiological characteristics, as well as on phylogenetic analysis of ITS, βtub, cox1, cox2 loci using Bayesian inference and the maximum likelihood method. This approach is widely used in this area of research and is beyond doubt. The design of the experiments was compiled correctly. The results are presented clearly and without any doubt. Overall, I recommend this manuscript for publication in this form.

 Overall, I recommend this manuscript for publication in this form.

Author Response

(The authors gave the same response as above.)

Reviewer 3 Report

Descript of new species are very important, I undersatand. 

Could you please consider my comments on attached PDF, and revise the manuscript to be accepted.

Author Response

(The authors gave the same response as above.)

Round 2

Reviewer 3 Report

The manuscript has been revised. However, unfortunately, I could find no response to some my previous comments.

I will comment again and some new comments below. I would like you to consider them.

l   As you mentioned in the Abstract, you identified these two new species based on morphological features as well as the phylogenetic analysis. However, result of “morphology” is missing in the “3. Results”. After showing the morphological and phylogenetical results, you can conclude those two groups should be new species. Could please consider them and re-construct the part of “Results”.

l   There are two morphological groups in Pythium banihashemianum. Could you please explain why you decide them as one species (intra-specific variation), instead of separate species.

P1 L13: I am not sure if physiological features are used for identification. Please consider it.

P4 L139:growth rate was tested on PDA…

 (Comment again) Based on the identification key (van der Plaats-Niterink), PCA is used for growth rate test, so it would be better to use PCA for this research. Please consider it.

P4 Pathogenicity

Could you please explain the natural occurrence of the disease in the collection area of your samples somewhere in the text.

P8 L243

Have you found rice disease in the field? It is important to explain whether the pre- and post-emergence damping-off, and crown rot are seen only at the experimental situation, or it is actually seen in field.

P9 L271, in figure legends of Fig. 4, 5: Group I à Morphotype I

in figure legends of Fig. 4, 5: Group II à Morphotype II

P15 449: the presence of an elongated clavate antheridium

 (Ask again) This is unclear in your photos. As your explanation, this feature is important to characterize of this species, so please show more clearer photos.

Table 3

 (Comment again) It is not necessary to describe data for all strains. Summarize them as characteristics of this species, including differences between morphotypes of P. banihashemianum. Table S6 would be better to include the manuscript. My suggestion is Table 3 is going to be a supplementary table, and Table S6 is going to be Table 3 instead.

Author Response

Manuscript “Pythium banihashemianum sp. nov. and Globisporangium izadpanahii sp. nov. two new oomycete species from rice paddies in Iran”; manuscript ID jof-2955950 by Salmaninezhad et al. submitted to Section “Fungi in Agriculture and Biotechnology”; Topical Collection “Advances in Plant Pathogenic Fungi: Diagnosis, Biological Control, and Eco-Sustainable Formulations” of Journal of Fungi

Answer to Editor and Reviewer n. 3

We acknowledge and value the time and effort invested in reviewing our manuscript. We addressed all the comments meticulously. The corresponding modifications and enhancements implemented in the revised manuscript are itemized in our responses below, typed in blue fonts.

Reviewer n. 3

  1. [In the related form] Does the introduction provide a comprehensive yet concise overview about the state of knowledge in the area of research? No.

A: We tried to cover every aspect of Pythium s. l. in the introduction section. First, we stated the importance of Pythium s. l. species in plant pathology and how they have an impact on different crops worldwide. We then discussed the history of Pythium s.l. phylogeny, tracing its evolution from its initial classification to its current categorization into new genera. We also included the major morphological features of each new genus and especially those related to the two main groups we were going to focus on our manuscript (i.e., Pythium s. s. and Globisporangium). Lastly, we brought up the importance of Pythium species in rice paddies worldwide with an emphasis on their distribution and identification in Iran. Hence, we believe we have covered everything required for the description of new taxa and their potential pathogenicity on the focused host (i.e. rice).

  1. As you mentioned in the Abstract, you identified these two new species based on morphological features as well as the phylogenetic analysis. However, result of “morphology” is missing in the “3. Results”. After showing the morphological and phylogenetical results, you can conclude those two groups should be new species. Could please consider them and re-construct the part of “Results”.

A: We considered the reviewer's comment and revised the first part of the 'Results' section, changing the title to "3.1. Morphology of isolates of Pythium (see line 184 of the revised manuscript) and addressing some general morphological features in this section. However, to avoid repetition, we provided a detailed explanation of the specific morphological features in the Taxonomy section. The modified text and corresponding Lines are reported here for the convenience of Editor and reviewer. See L. 192-197 of the revised manuscript: “Significant morphological differences were observed in the sexual structures of the examined isolates, distinguishing them from other known species. Despite these major morphological distinctions, molecular similarity in the phylogenetic analyses made it possible to distinguish the first group of isolates being into two different subgroups based on morphological features (see the Taxonomy section).” See also L275-278: “Although no differences were observed in the sequences of isolates assigned to this species, their morphological characteristics were distinct, leading to their division into two morphological groups. Morphotype I:”

See L. 295-296 of the revised manuscript: “Comparison of morphological characteristics of the isolates assigned to P. banihashemi-anum sp. nov. morphotype I with its sister taxa are shown in Table 3. Morphometric features of the examined isolates are shown in Table S6.”

See L.362-364 of the revised manuscript: “Comparison of morphological characteristics of the isolates assigned to P. banihashemianum sp. nov. morphotype II with its sister taxa are shown in Table 3. Morphometric features of the examined isolates are shown in Table S6.”

  1. There are two morphological groups in Pythium banihashemianum. Could you please explain why you decide them as one species (intra-specific variation), instead of separate species.

A: We addressed the issue of the reviewer explaining why we distinguished the two groups.  Please see P5 L. 192-197 of the revised manuscript: “Significant morphological differences were observed in the sexual structures of the examined isolates, distinguishing them from other known species. Despite these major morphological distinctions, molecular similarity in the phylogenetic analyses made it possible to distinguish the first group of isolates being into two different subgroups based on morphological features (see the Taxonomy section).

See also P9: L272-277 of the revised manuscript “Although no differences were observed in the sequences of isolates assigned to this species, their morphological characteristics were distinct, leading to their division into two morphological groups.”

  1. P1 L13: I am not sure if physiological features are used for identification. Please consider it.

A: As mentioned in our previous statements, physiological features, such as daily growth rate and growth on different media, optimum temperature for radial mycelium growth, are considered important characteristics in species description. These features are recognized in taxonomic studies, including van der Pläats-Niterink (1981), Dick (1990), Tojo et al. (2012), and several other publications.

  1. P4 L139:growth rate was tested on PDA…(Comment again) Based on the identification key (van der Plaats-Niterink), PCA is used for growth rate test, so it would be better to use PCA for this research. Please consider it.

A: Changed accordingly. It was just a spelling mistake.

  1. Could you please explain the natural occurrence of the disease in the collection area of your samples somewhere in the text.

A: The text revised accordingly. Please see P4: L165-167 of the revised manuscript “2.4 Pathogenicity

With the exception of the Persepolis region (Table 1), pre- and post-emergence damping off, root rot, and in rare cases, crown rot were observed in 95% of the plants sampled from the selected regions.”

  1. P8 L243 Have you found rice disease in the field? It is important to explain whether the pre- and post-emergence damping-off, and crown rot are seen only at the experimental situation, or it is actually seen in field.

A: Changed accordingly. Please see P4: L L165-167 (above reported sentence) .

  1. P9 L271, in figure legends of Fig. 4, 5: Group I à Morphotype I

in figure legends of Fig. 4, 5: Group II à Morphotype II

A: Changed accordingly.

  1. P15 449: the presence of an elongated clavate antheridium(Ask again) This is unclear in your photos. As your explanation, this feature is important to characterize of this species, so please show more clearer photos.

A: The elongated antheridium is clearly visible in Fig. 8-b. The existence of the elongated antheridium is only one of the characteristics of this species. This feature, along with other distinguishing characteristics, helps to identify this species. These include a perfectly plerotic oospore, globose hyphal swellings, lacking zoospore production, and most importantly, the presence of papillae on oogonium. These combined features differentiate this species from other Globisporangium spp.

  1. Table 3(Comment again) It is not necessary to describe data for all strains. Summarize them as characteristics of this species, including differences between morphotypes of  banihashemianum. Table S6 would be better to include the manuscript. My suggestion is Table 3 is going to be a supplementary table, and Table S6 is going to be Table 3 instead.

A: Changed accordingly.

Academic Editor Comments

  1. Like many other fungal and oomycete organisms, new species will be identified on a regular basis with the advent of new molecular tools. However, many readers may not be familiar with the rules of nomenclature. I suggest adding a short section in the discussion on the taxonomic aspect and significance of the discovery of new species.

A: The Discussion section revised accordingly. Please see the last paragraph of the Discussion section, L536-549: “The discovery of two new species, G. izadpanahii and P. banihashemianum, provides new insight into the diversity of oomycetes. The advent of molecular techniques and multigene phylogenetic analysis have greatly contributed to the advancement of the systematics of oomycetes and are valuable tools to set up a sound internationally rec-ognized taxonomic framework. Numerous new cryptic species were separated from species complexes and polyphyletic genera, like Pythium s.l., were split into clearly separate genera, differing also in morphological and physiological features. However, a putative new taxon cannot be regarded as a species until its hierarchical status has been determined and its name has been formally designated according to the rules of the International Code of Nomenclature for algae, fungi, and plants. The precise identification of a taxon has also phytopathological implications and is crucial for the effective management of plant diseases caused by oomycetes. For instance, of the two new species described in this study, both associated to symptomatic rice seedlings, only P. banihashemianum was proved to be pathogenic.”

Round 3

Reviewer 3 Report

The manuscript has been well revised according to my comments.

I have one question about PCA.  In the P3 L137, PCA has been prepared by potato extract, carrot extract, dextrose and agar. However, in van der Plaats-Niterink, dextrose is not included in the PCA. This should make big differnce of growth. So, plesase re-examin and change the data.

Also, in the Table3, there are information of some species. I wonder where some data are coming from, and Daily growth rate is on which medium from. For example, in the description of P. kashimirense, daily growth rate is 15 mm, and narrow chrysanthemum pattern on PCA, which are different with those of your manuscript. So, could you please reconfirm all data on Table3, and recheck the citation.

Author Response

Response to Reviewer 3

We sincerely appreciate your valuable comments and suggestions on our manuscript. Your insights have significantly enhanced the quality of our work.

  1. I have one question about PCA.  In the P3 L137, PCA has been prepared by potato extract, carrot extract, dextrose and agar. However, in van der Plaats-Niterink, dextrose is not included in the PCA. This should make big differnce of growth. So, plesase re-examin and change the data.

A: We have made changes according to your comment and cited the right reference. Changes are highlighted in light-blue.

  1. Also, in the Table3, there are information of some species. I wonder where some data are coming from, and Daily growth rate is on which medium from. For example, in the description of P. kashimirense, daily growth rate is 15 mm, and narrow chrysanthemum pattern on PCA, which are different with those of your manuscript. So, could you please reconfirm all data on Table3, and recheck the citation.

A: We would like to express our gratitude for mentioning this mistake that was not evidenced in previous revision round. The comparison table has now been re-checked, and the changes are highlighted in light-blue.

We have also submitted the text to a professional English mother-tongue scientific lecturer for further editing of the text. Moreover, we added a reference to improve the part of the Discussion which was added after the previous revision round.